# Description of Release Process of Doxorubicin from Modified Carbon Nanotubes

**DOI:** 10.3390/ijms222112003

**Published:** 2021-11-05

**Authors:** Dorota Chudoba, Monika Jażdżewska, Katarzyna Łudzik, Sebastian Wołoszczuk, Ewa Juszyńska-Gałązka, Mikołaj Kościński

**Affiliations:** 1Faculty of Physics, Adam Mickiewicz University, 61-614 Poznan, Poland; mojaz@amu.edu.pl (M.J.); sebastian.woloszczuk@amu.edu.pl (S.W.); 2Frank Laboratory of Neutron Physics, Joint Institute for Nuclear Research, 141980 Dubna, Russia; katarzyna.ludzik@chemia.uni.lodz.pl; 3Department of Physical Chemistry, University of Lodz, 90-236 Lodz, Poland; 4Institute of Nuclear Physics Polish Academy of Sciences, 31-342 Kraków, Poland; ewa.juszynska-galazka@ifj.edu.pl; 5Research Center for Thermal and Entropic Science, Graduate School of Science, Osaka University, Osaka 560-0043, Japan; 6Department of Physics and Biophysics, Faculty of Food Science and Nutrition, University of Life Sciences, 60-637 Poznan, Poland; mikolaj.koscinski@amu.edu.pl; 7NanoBioMedical Centre, Adam Mickiewicz University, 61-614 Poznan, Poland

**Keywords:** modified carbon nanotubes, drug delivery, doxorubicin, kinetics of release, drug release mechanism

## Abstract

The article discusses the release process of doxorubicin hydrochloride (DOX) from multi-wall carbon nanotubes (MWCNTs). The studies described a probable mechanism of release and actions between the surface of functionalized MWCNTs and anticancer drugs. The surface of carbon nanotubes (CNTs) has been modified via treatment in nitric acid to optimize the adsorption and release process. The modification efficiency and physicochemical properties of the MWCNTs+DOX system were analyzed by using SEM, TEM, EDS, FTIR, Raman Spectroscopy and UV-Vis methods. Based on computer simulations at pH 7.4 and the experiment at pH 5.4, the kinetics and the mechanism of DOX release from MWNT were discussed. It has been experimentally observed that the acidic pH (5.4) is appropriate for the efficient release of the drug from CNTs. It was noted that under acidic pH conditions, which is typical for the tumour microenvironment almost 90% of the drug was released in a relatively short time. The kinetics models based on different mathematical functions were used to describe the release mechanism of drugs from MWCNTs. Our studies indicated that the best fit of experimental kinetic curves of release has been observed for the Power-law model and the fitted parameters suggest that the drug release mechanism of DOX from MWCNTs is controlled by Fickian diffusion. Molecular dynamics simulations, on the other hand, have shown that in a neutral pH solution, which is close to the blood pH, the release process does not occur keeping the aggregation level constant. The presented studies have shown that MWCNTs are promising carriers of anticancer drugs that, depending on the surface modification, can exhibit different adsorption mechanisms and release.

## 1. Introduction

Carbon nanomaterials especially carbon nanotubes (CNTs) which are characterized by unique properties, including large surface area, high size stability and the ability to wall functionalization [1,2]. Surface modification of nanotubes fenhances their application as potential drug carriers. However, CNTs are hydrophobic with low solubility in the common solvents which can limit their application in biomedicine. It has been found that the functionalization of CNTs surfaces makes them more soluble in the biological environment and allows them to efficiently attach the functional molecules [3,4]. It has been also shown that modified CNTs exhibit a reduction of cytotoxicity and improve biocompatibility [5,6].

Through appropriate functionalization, CNTs can be successfully used as nanocarriers to transport anticancer drugs such as Doxorubicin (DOX), anthracycline antibiotic with antineoplastic activity used to treat a wide spectrum of cancer diseases including breast, bladder, stomach, bone and neural tumours. However, as with other cytostatic, the use of DOX causes several toxic side effects such as cardiotoxicity, alopecia, vomiting, leucopenia, and stomatitis, limiting the successful application of this drug in chemotherapy. The reduction of the undesirable impact of DOX in cancer therapy without reducing therapeutic effects can be obtained by adopting new solutions based on carbon nanocarriers.

The crucial issue in drug carrier design consists of two steps: achieving successful drug adsorption onto the nanocarriers and efficiently releasing the drug into the tumour. Izadyar et al. [7] have shown that the efficiency of adsorption of modified single-wall carbon nanotubes (*m*SWCNTs) is higher than that observed for pristine *p*SWCNTs. They have also noticed that the adsorption process of DOX onto CNTs is relatively difficult and can be improved by extension of the functional groups on the carbon surface. Matyszewska et al. present studies of different modes of carbon surface modification of systems *m*SWCNTs with DOX and their behavior with model biological membrane [8]. It was found that side modification of SWCNTs is more efficient and allows for more efficient conjugation of the drug to CNTs compared to the modification of the end of nanotubes. Zhang et al. [9] noted that by manipulating the surface potentials of the modified nanotubes, the loading efficiency and release rate of the associated doxorubicin could be controlled. The drug delivery system should respond to the physiologically acidic pH since the tumour microenvironment is acidic. Wang et al. [10] have shown that the desorption of DOX from oxidized MWCNTs hardly occurs while incubated in buffer solution at pH 7.4 and pH 5.5 in contrast to lower pH, which benefits the desorption. Farahani et al. [11] have remarked that the release process of DOX from MWCNTs-PEG was pH-dependent, and the release rate was much higher at pH 5.5 than that at pH 7.4. The preferential DOX release in an acidic pH rather than in neutral pH was also noted by Chen et al. [12]. What is more, it was shown that polymer-DOX nanoconjugates significantly improved the effectiveness of anti-tumor of drug and alleviated it side effects. The influence of pH, surface charge, and DOX-loading within the self-assembled nanoparticles on release and stimulation of DOX-induced cardiotoxicity were also studied [13]. It was indicated that noncovalently DOX encapsulated nanoparticles induce cardiotoxicity in primary cardiomyocytes, while covalent conjugation of DOX to a pH-sensitive nanocarriers reduced the cardiotoxicity. The efficacy of loading and releasing of DOX using CNTs in various pH conditions was also studies by MD simulations [14]. It was demonstrated that the bonds between MWCNTs and DOX was stronger and the release process of DOX was slower in the cancer environment compared to SWCNTs.

Undoubtedly, the efficiency of drug release from nanocarriers is a major determinant of its biological use. Therefore, the evaluation of drug release kinetics is one of the most important research factors. The efficiency of the release process depends on several factors, including the solute and nanocarriers properties, the drug solubility, interaction drug—matrix and the release environment. Although studies of adsorption and release processes of DOX—CNTs complex have been reported [7,9,10,11,15,16,17,18,19], there is little known about the kinetics mechanism of these processes. The current study gives a deep insight not only into the kinetics of release but also into the release mechanism of the drug.

In this paper, we present the Doxorubicin hydrochloride (DOX) release process from the modified MWCNTs based on experimental and molecular simulations results. Experimental research were carried out at acidic pH, which is typical for the tumor microenvironments. The release of DOX from CNTs under neutral pH (blood pH) conditions was analyzed using MD simulations. In addition, based on the research results, the DOX release mechanism was also analyzed. The mechanism of release has been evaluated by means of different kinetics models.

## 2. Results and Discussion

### 2.1. Fourier Transform Infrared (FTIR) Spectroscopy

FTIR measurements were carried out on pristine *p*MWCNTs and modified *m*MWCNTs as well as on pure DOX, and DOX adsorbed onto modified MWCNTs. The obtained results are shown in Figure 1. In the case of *m*MWCNTs (a), the FTIR spectrum shows two characteristic peaks, one—at 3424 cm^−1^ and 1424 cm^−1^ which indicates O-H stretching vibration hydrogen groups, deformational COH vibration at 1091 cm^−1^, 1047 cm^−1^, 632 cm^−1^ and the second peak—at 1628 cm^−1^ attributed to C=O stretching vibration groups observed.

The obtained result can suggest that the surface of *m*MWCNTs contains hydroxyl and carbonyl groups introduced by oxidation both during the production process and surface modification. The comparison of *p*MWCNTs and *m*MWCNTs has shown that the intensities of the vibrational band of both samples are different. The band positions of pristine and modified MWCNTs are also shifted relatively each other. These results confirmed that the studied systems are various. The comparison spectra of doxorubicin and DOX spectra adsorbed onto chemical modified MWCNTs—see Figure 1b presents the FTIR spectrum of pure DOX. This spectrum shows several characteristic vibrational bands. For DOX in *m*MWCNTs, one can find a shift of positions of vibrational bands in comparison with the spectra obtained for *m*MWCNTs. One can conclude that DOX and *m*MWCNTs interact with each other. The bands of characteristic atomic groups of the DOX molecule were observed, namely at 1100 cm^−1^ the visible band corresponds to the C-N stretching vibration, as well as in the range of stretching vibrations (4000–2500 cm^−1^) the band at 3400 cm^−1^ is assigned to vibrations of N-H bond, and C-H at 2931 cm^−1^. The range between 3250 cm^−1^ and 2750 cm^−1^ is due to the stretching bands of the C–H groups which indicate the presence of alkane groups. The peak at 1732 cm^−1^ is related to the stretching bands of the C=O groups and indicates carboxyl groups. Another band visible in the spectrum at 1586 cm^−1^ is assigned to the anthraquinone ring, whereas the peak at 1413 cm^−1^ is due to the stretching bands of the C–C groups [20]. The peak at 1285 cm^−1^ is due to the stretching vibration bands of the C–N groups, 1071 cm^−1^ peak shows the presence of the stretching bands of the C=O groups and the peak at 991 cm^−1^ illustrates the stretching bands of C−O groups. The spectrum obtained for DOX onto *m*MWCNTs is given in Figure 1b. The result clearly confirms the presence of DOX onto *m*MWCNTs. The broadband at 3328 cm^−1^ can be attributed to the stretching vibration of O-H groups and the band at 1732 cm^−1^ is due to C=O asymmetric stretching vibration groups. These two peaks are related with the presence of *m*MWCNTs in the system. The other bands at 1641 cm^−1^, 1279 cm^−1^, 1001 cm^−1^ which are visible in the spectrum could be considered as the characteristic peaks of DOX. After adsorption of DOX onto *m*MWCNTs the shift of bands to the lower or higher frequency is observed. It is worth noticing that in case-NH_2_ groups, the position and shape of the band depend on the degree of involvement of this group through N-H bonds in hydrogen bonding. The presence of the hydroxyl group makes it difficult to identify the amino group bands as these vibrations overlap. The hydrogen bond causes that the band corresponding to the NH bond stretching vibrations shifts towards lower wavenumber. In the spectra containing the amino groups, there are two sharp medium-intensity absorption bands corresponding to the asymmetric (about 3500 cm^−1^) and symmetrical (about 3400 cm^−1^) stretching vibrations of the free amino group -NH_2_. Thus, in the spectra simultaneously bands corresponding to the vibrations of the bound and free groups with the participation of N-H bonds can be observed. Individual vibrational bands of the C=O or N-H groups are indicated, Figure 1.

The change of position and intensity of vibration bands depicting the movement of functional groups indicate participation in the formation of intermolecular bonds. The shift to higher wavenumber by about 20–40 cm^−1^, compared to groups not involved in the interaction is typical. This result can indicate the participation of functional groups in DOX bonding as well as DOX—DOX interactions by hydrogen bonds that represent the majority.

### 2.2. Raman Spectroscopy

The Raman spectroscopy is one of the most useful methods to characterize carbon materials because this technique provides important information about the purity andstructure of the above nanomaterials. The application of SERS can enhance the Raman signals of target species adsorbed on Ag or Au metallic substrates by as much as 6 to 14 orders of magnitude giving an opportunity to detect single molecules. With these excellent advantages, SERS is suitable to determine doxorubicin. Figure 2 shows the Raman spectra obtained at room temperature using Renishaw software of *p*MWCNTs (a), *m*MWCNTs (b), doxorubicin (c—633 nm laser wavelength and c’-785 nm laser wavelength) and DOX adsorbed onto *m*MWCNTs (d). The three typical graphite bands are observed in the Raman spectra of pristine and modified MWCNTs. The band visible at about 1340 cm^−1^ (D band) is assigned to defects and disorders of sp2 hybridized carbon systems. The band at about 1580 cm^−1^ (G band) is related to the tangential C–C stretching mode in graphitic materials. The second-order mode of the D-band shown in the Raman Spectra, is a band at about 2670 cm^−1^ (2D band) [21,22]. The Raman Spectra of MWCNTs also demonstrate weak bands at 2910 cm^−1^ (D + G band) and at 3240 cm^−1^ (2D’ band) which are caused by lattice disorder in the structure of the nanotubes [23,24].

The ratio of the intensity of D peak to the intensity of G peak ID/IG has been determined on the data obtained from the Raman measurements. The ratio of these intensities is related to the number of defects in the carbon material [25]. The calculated ratio of the ID/IG intensities of the both analyzed samples is about 0.41, which means that the modification process has not influenced the increase of the amount of the defects in the system. In the both cases, pristine and modified MWCNTs, the intensity of the G band is much higher than the intensity of the D band, which indicates a small number of defects in the carbon structure. The similar intensity of these two bands corresponds to a large number of structural defects. The bands visible in the Raman spectrum shown in Figure 2c’ are associated with doxorubicin [26,27]. The peaks at 1446 cm^−1^ and 1575 cm^−1^ are attributed to the skeletal ring vibrations; the peaks between 1200 cm^−1^ and 1300 cm^−1^ could be the results of bonding motions C–O, C–O–H and C–H, respectively. The band at 1086 cm^−1^ can be attributed to the stretching vibration of the C=O groups. The peaks visible at 352 cm^−1^ and 450 cm^−1^ are related to C–C–O and C=O in-plane deformation. The Raman spectrum, which is given in Figure 2d, clearly shows the evidence of the presence of doxorubicin onto *m*MWCNTs in this case, except for the spectrum characteristic for pure doxorubicin illustrated in Figure 2c. The bands typical for carbon materials (D, G and 2D) are also observed.

### 2.3. Microscopic Techniques

#### 2.3.1. SEM Analysis

The morphological and elemental studies of analyzed samples were performed using SEM/EDS technique. The morphology of the studied materials, which resemble a spaghetti-like shape, is demonstrated in Figure 3. The obtained results have shown a typical morphology for carbon nanotubes. In addition, it should be highlighted that the adsorbed doxorubicin causes formation of objects consisting of nanotubes sticking 178 together. This finding suggests that doxorubicin molecules exist in the space between MWCNTs.

The qualitative and quantitative determination of surface composition via EDS was also applied. The obtained results are presented in Figure 4. EDS analysis has revealed the presence of carbon, nitrogen, oxygen and a small amount of chlorine. As it can be seen, the chemical modification of MWCNTs caused increasing the content of nitrogen and oxygen after treatment with nitric acid. The content of oxygen after MWCNTs modification is about 2% higher than before the modification process. The system composite with DOX adsorbed onto MWCNTs has increased the content of nitrogen, oxygen, chlorine. These elements are typical for doxorubicin hydrochloride. The obtained result has clearly confirmed that DOX was adsorbed onto MWCNTs.

#### 2.3.2. TEM Analysis

Transmission Electron Microscopy was used to study the structural characteristics of the DOX–MWCNTs complex. The TEM images in Figure 5 show that the morphology of the materials is very similar. It important to mention that the chemical modification of MWCNTs is confirmed by specific shapes of etched CNTs surfaces. Furthermore, the adsorption of doxorubicin caused the formation of objects consisting of nanotubes sticking together.

### 2.4. UV-VIS Spectroscopy

The spectrophotometric standard curve method was used to determine the doxorubicin concentration of the both adsorption and release processes. The maximum adsorption capacity obtained for *m*MWCNTs was ca. 3800 mg/g, while in the case of *p*MWCNTs, it was ca. 175 mg/g [28]. The adsorption process of DOX onto MWCNTs was performed by means of a buffer of neutral pH (about 7.4). In neutral pH, the carboxylic groups on MWCNTs surface are negatively charged and can better adsorb the aromatic and amine groups of the doxorubicin [29]. In order to investigate the release process of DOX from *m*MWCNTs, several tests were performed using different volumes of acetic buffer. The release of DOX is determined by factors such as temperature or pH [30,31]. Generally, increasing temperatures from room temperature (298 K) to body temperature leads to higher rate of DOX release from different type of careers [32]. The pH value influences on hydrogen bonding strength created between -NH_2_ and -OH of the DOX as well as –COOH and –OH of the nanotubes [33]. As a results acidic conditions weaken hydrogen bond which in turn causes higher release rate. What is more acidic pH is typical for the tumor microenvironments. Thus, in presented manuscript the experiments were carried out for pH = 5.4 at 298 K.

It has been observed that the concentration of released doxorubicin was different for varied volumes of acetic buffer. The highest release value was observed in the case of 50–100 mL of the acetic buffer. The obtained results are presented in Table 1.

The spectrophotometric standard curve method was also applied to determine the kinetics of release of doxorubicin from *m*MWCNTs. The release process was determined for the volume of the acetic medium, which was 50 mL, and the obtained result is given in Figure 6. During the first 2 min, the release process is very fast (about 70% of DOX is released), while in the following minutes, this process is more stable and much slower.

The rapid release process at the beginning can be related to molecules of DOX, which formed agglomerates not related to MWCNTs or located as the outside layers of DOX adsorbed in carbon walls. On the other hand, the gradual release in the following minutes may be associated with the release of DOX molecules from inner and contact layers as well as from inside the carbon tube. Based on the obtained results, it can be said that the equilibrium is reached after 20 min. After this time, almost 90% of DOX has been released from *m*MWCNTs. There is no doubt that pH, temperature, surface functionalization, type of drug and the properties of the nanocarrier determine kinetics of drug release. Taking into account the rate of release presented in literature, observed in our case kinetics of DOX release is very fast [34,35,36].

#### Mechanism of the Drug Release

Understanding of the release process is very important in the control of drug release. To describe the release mechanism, the kinetics models based on different mathematical functions can be used. To examine the obtained kinetic release profile, the following models were applied: zero-order, first-order and second-order kinetics, Higuchi, Hixson–Crowell, Weibull, and Korsmeyer–Peppas/Ritger-Peppas (Power-law model).

The zero-order kinetics model [37] can be expressed by the following equation:*C_t_* = *C*_0_ + *K*_0_*t*
(1)
where *C_t_* is the amount of the drug released over time *t*, *C*_0_ is the amount of the drug in solution before release (generally *C*_0_ = 0), and *K*_0_ is the zero-order release rate constant.

This model defines the process which takes place at a constant rate independent of active agent concentration.

The first-order kinetics model [38] is as follows:*l*(*C_d_* − *C_t_*) = *ln*(*C_d_*) – *K*_1_*t*(2)
where *C_t_* is the amount of the drug released over time *t*, *C_d_* is the amount of the drug before dissolution and *K*_1_ in the equation is the first-order release rate constant. The first-order kinetics model describes the release process whose rate is proportional to the concentration of the drug in the system.

The second-order kinetics model [38] can be represented by the equation:(3)$$\frac{1}{\left(C_{d}-C_{t}\right)}=\frac{1}{C_{d}}-K_{2}t$$
where *C_t_* is the amount of the drug released over time *t*, *C_d_* is the initial amount of drug (amount of drug remaining at time 0), and *K*_2_ is the second-order rate constant.

Higuchi model [39] is expressed as:(4)$$C_{t}=K_{H}t^{\frac{1}{2}}$$
where *C_t_* is the amount of the drug released over time *t* and *K_H_* is the Higuchi rate constant.

Hixson-Crowell model [40] is described by the following equation:(5)$$C_{d}^{\frac{1}{3}}-C_{l}^{\frac{1}{3}}=K_{H-C}t$$
where *C_d_* is the initial amount of drug, *C_l_* is the amount of drug left in the formulation over time *t*, and *K_H-C_* is the Hixon-Crowell rate constant.

Weibull model [41] can be given by the equation:(6)$$C_{m}=C_{s}\left[1-e^{-\frac{{\left(t-T\right)}^{b}}{a}}\right]$$
where *C_m_* is the amount of drug dissolved as a function of time *t*, *C_s_* is the total amount of drug being released, T means the latency time of the release process, *a* is the scale parameter which defines the timescale of the process, and *b* characterizes the type of curve (for *b* = 1 the shape of the curve corresponds to exponential, for *b* > 1 the shape of the curve is sigmoid and for *b* < 1 the curve is parabolic).

Korsmeyer–Peppas and Ritger-Peppas model (Power-law model) [42,43] can be simplified as follows:(7)$$C=\frac{C_{t}}{C_{\infty }}=Kt^{n}$$
where *C* is the amount of drug released, *C_t_* is the amount of drug released over time *t*, $C_{\infty }$ is the amount of drug at the equilibrium state, *K* means the constant of incorporation of 283 structural modifications and geometrical characteristics of the system, and parameter *n* is the exponent of release related to the drug release mechanism, in the function of time *t*.

In the present work, several kinetic models have been used to fit the experimental data to the theoretical curve. First, the kinetic release parameters are calculated from various kinetic models, such as zero-, first-, and second-order kinetics, as well as the Higuchi, Hixson–Crowell, Weibull and Power-law model, which is listed in Table 2.

The criterion for the most suitable model fitted to the experimental data was based on the high degree of correlation coefficient of the DOX release profile. The zero- and first- order same as Hixson-Crowell models were completely unsuitable for describing the release of DOX from MWCNTs. In the case of these models, the values of the correlation coefficient R^2^ indicating the appropriate kinetic model were less than 0.6. The Hixon-Crowell model was also inappropriate to describe the release mechanism of drugs from carbon nanotubes. Evaluation of different kinetic models for DOX release from MWCNTs indicated that three models, i.e., the second-order, Weibull and Power-law models were more appropriate than the others due to a better coefficient of correlation which was closer to unity. Among these three kinetic models, the Power-law model was more appropriate than the other two models because of the smallest differences between the experimental curve and the release profile obtained. For the Power-law model, the coefficient value, R^2^, was close to 1 (R^2^ ≈ 0.98) indicating that this model is pretty good for fitting the release mechanism of DOX from MWCNTs. The graphical presentation of the Power-law model, which properly describes the release mechanism of DOX from MWCNTs, is given in Figure 7.

The Power-law model allowed one to characterize the drug release mechanism by analyzing the diffusion exponent *n*. According to this model, the value of the *n* parameter determined the mechanism of drug release: when the value of *n* is below 0.5, the drug release is governed by the Fickian mechanism; the non-Fickian model is observed if *n* is between 0.5 to 1 and the mechanism of drug release is governed by diffusion and swelling; the model is also non-Fickian (Case II) when *n* = 1 and the Super Case II model is characterized when *n* > 1.

The *n* exponent value (Table 2) obtained for the studied system is smaller than 0.5, which indicates that DOX release from MWCNTs is governed by the Fickian diffusion mechanism. As it is shown above, the experimental data were also fitted using the Weitbull model, for which the correlation coefficient was close to 0.9. Using this model, the shape factor—*b*, (Equation (6)) was determined. The *b* parameter is useful to indicate the type of mechanism of the drug transport. It is known that the value of *b* less or equal to 0.75 indicates Fickian diffusion, while the value between 0.75 and 1 is related to combined mechanisms such as Fickian diffusion and swelling-controlled transport. For values of *b*, which is higher than 1, the complex release mechanism is observed [44]. Therefore, the obtained value of *b* (Table 2) indicates that the drug release mechanism was controlled by Fickian diffusion. This result is consistent with the one obtained by using the Power-law model.

The understanding of drug release mechanisms from nanocarriers is very important for the potential application of these systems in the treatment of tumours. This mechanism depends on many factors, including the kind of drug and matrix, temperature, pH, and environmental conditions. There are many studies of the analysis of drug release mechanisms from nanocarriers described in the literature. P. Soares et al. [45] studied the release process of DOX at different pH from two drug delivery systems based on chitosan nanoparticles. They have noted that the mechanism of DOX release is anomalous or mixed between Fickian diffusion and relaxation transport. M. Fallahi-Samberan et al. [46,47] investigated the kinetics of DOX release from composite particles by different mathematical models. They have found that the kinetics release of DOX was well described by the Korsmeyer–Peppas model, which means that the controlling factor in the release process was a polymeric matrix. The mechanism of DOX release was also investigated by P. Abasian et al. [48]. They showed that the process of DOX release from nanofibers (chitosan/PLA/NaX zeolite and chitosan/PLA/NaX zeolite/ferrite) was well described by the Krosmeyer-Peppas kinetic model, and based on kinetics parameters. They noted that the mechanism of DOX release was related to Fickian diffusion. The existence of the Fickian diffusion mechanism was also found in the case of drugs release from MWCNTs [49]. The study of the release mechanism of cytostatic drugs from MWCNTs functionalized by polymers was also carried out [50]. As it was shown, depending on the kind of drug, pH and temperature conditions, the proper fit of release profiles was obtained by using different mathematical kinetic models. These results can indicate different drug release mechanisms. A different mechanism of drug release depending on the pH conditions was also observed for drug carriers such as Carbon Nanotubes Hydrogel [51]. It has been found that the release mechanism in neutral pH is related to the non-Fickian diffusion process, which involves surface and corrosion diffusion, while in lower pH, the Fickian diffusion was observed.

### 2.5. Molecular Dynamics Simulations

Simulations were performed for both MWCNTs + DOX and SWCNTs + DOX systems. In each of the two cases, the study of the release process under neutral pH conditions required first preparing the input system, i.e., simulating the adsorption process of doxorubicin molecules on the nanotube until saturation, followed by simulating the system in neutral pH solution and observing the degree of resorption.

For this reason, the simulations were divided into two parts. In the first part, we led to the adsorption of DOX molecules on a given type of nanotube under in vacuo conditions. In the second part, we added water to such a saturated (CNT + DOX) system to study the instability of the aggregates at neutral pH. To do this, first, we placed the given nanotube centrally in the simulation box along its *y*-axis. Then we added DOX molecules in the amount which satisfied the mass ratio condition (mDOX/mCNTs) = 4/1. Next, we minimized the energy to get rid of all unphysical positions of the molecules. This process was continued until the maximum force in the system reached the value below 1 kJ mol^−1^ nm^−1^.

We will first focus on the analysis of the results obtained for MWCNTs. The system with MWCNTs prepared according to the above procedure is seen in Figure 8. At this point, we started the simulation in the NVT ensemble and simulated the system for 20 ns making sure that the adsorption and aggregation process had finished, i.e., the observed parameters such as the size of the aggregates or their number did not change over time.

We then added *t4tip* water to the system, reran the energy minimization process to exclude any non-physical position of the water molecules, and then started the simulation in a pH-neutral environment for 15 ns (total simulation time 35 ns). The output system from simulation with water is shown in Figure 9.

In Figure 10a, we see the number of aggregates (grey squares) and average aggregate size (red circles) as a function of the total simulation time, in vacuo and with water, respectively. The number of aggregates decreases rapidly with time from 662 to 4 in the first 2.5 ns, while the average aggregate size increases from 10.7 to 1131 in the same time.

If we look at Figure 10b, we can see the total percentage of DOX molecules adsorbed by MWCNTs (grey squares) and their percentage adsorbed inside the nanotube (red circles). It can be clearly seen that the initially small values of both of these quantities increase with time, and similarly, after about 2.5 ns, they reach a plateau, 72.9% for DOX molecules adsorbed by MWCNTs and 4.2% for DOX molecules adsorbed inside the nanotube.

For all four physical quantities, we see that this state no longer changes until the end of the in vacuo simulation (*t* = 20 ns) as well as after placing the system in neutral pH water environment (between *t* = 20 ns and *t* = 35 ns), which is very close to the pH characteristic of blood. We would like to point out the small fluctuations in the proportion of DOX molecules adsorbed inside the nanotube. The reason for these fluctuations is the algorithm used in counting these molecules. This algorithm divides the nanotube into single carbon rings of diameter *f* = 6.42 nm arranged parallel to each other (coaxially). For each ring its center of mass is determined and from this center of mass a virtual sphere of radius *r* = 6.42 nm is made. Then, those DOX particles whose center of mass lies inside the given sphere are assigned to the given ring. Finally, duplicates are removed from the sum of DOX sets for all analyzed rings, which de facto reduces the sphere analysis to analysis inside narrow cylinders coaxial with the nanotube. This method implicitly assumes that the nanotube maintains perfect cylindrical symmetry, which is not true because in reality, nanotubes exhibit some longitudinal and transverse flexibility. This is the reason for the small fluctuations (red circles) in Figure 10b. The relatively small amount of adsorbed molecules inside the nanotube is in turn explained by the fact that DOX molecules are not only adsorbed by the nanotube, but also form free monomolecular aggregates in the meantime. Therefore, the number of clusters (grey squares) in Figure 10a does not drop to unity but remains at the level of a few. In practice, for very long times we may suspect that free large DOX aggregates may attach to the large MWCNTs + DOX aggregate in contact, however, for times achievable in molecular simulations we have not observed that.

To summarize the MWCNTs section, we see that the neutral pH (water) environment does not affect the aggregated MWCNTs + DOX system.

For the SWCNTs + DOX system, the procedure for preparation, energy minimization, and the simulations themselves were analogous except that in this case the simulation time in aqueous solution was extended to 60 ns, resulting in the total simulation time *in vacuo* + with water of *t* = 80 ns. We have chosen the single-wall nanotube for such a long simulation due to the lower computational complexity of the overall simulation. In Figure 11a, we see the time dependence of the number of aggregates (grey squares) and the average aggregate size (red circles). The number of aggregates decreases rapidly with time from 872 to 4, while the average cluster size increases rapidly from 2.7 to 354 after time *t* = 7.5 ns. Similarly, in Figure 11b, we see that in the first 7.5 ns of the simulation, the total percentage of the adsorption process of DOX on the nanotube (grey squares), as well as the percentage of adsorption inside the nanotube (red circles), saturates to the values of 31.5% and 1.6%, respectively.

All four parameters analyzed in Figure 11 show a *plateau* after this time, both in vacuo (between time *t* = 0 ns and *t* = 20 ns) and water solvated systems (between time *t* = 20 ns and *t* = 80 ns). As in the case of MWCNTs + DOX, the proportion of adsorbed DOX molecules inside SWCNTs shows minor fluctuations and, as in the previous case, the algorithm of counting these molecules is responsible for this, which assumes perfect cylindrical symmetry of the nanotube when the nanotube exhibits some longitudinal and transverse flexibility. It is worth noting that these fluctuations, as in the case of the MWCNTs + DOX system, reduce in the system with water, indicating that the aqueous environment stabilizes slightly the cylindrical symmetry of the nanotubes. These calculations have shown that increasing the simulation time several times does not change the observed system parameters in the system with water. As in the case of MWCNTs + DOX, we can assume that on significantly longer time scales free DOX aggregates may or may not be adsorbed when in contact with the SWCNT + DOX cluster. It is also worth noting by comparing Figure 10 and Figure 11 that the adsorption process on the nanotube is significantly faster in the case of MWCNTs. The stability of the percentage of DOX adsorption on the inner part of the nanotubes (Figure 10b and Figure 11b—red circles), in turn, shows that in the confined spaces the presence of water does not change in the time ranges we simulated.

The results obtained for both systems in the simulated time ranges have shown that in the presence of a neutral pH environment all parameters related to the adsorption of DOX molecules on the simulated nanotubes and their free aggregation do not change. This indicates the stability of the above systems at neutral pH in both open and confined spaces. Moreover, the obtained results indicate that water solvent stabilizes slightly the cylindrical symmetry of the nanotubes, i.e., their symmetry remains more cylindrical. In summary, the aqueous pH environment does not affect the aggregated CNTs + DOX system, which is in general agreement with [10] and allows concluding that in an environment close to blood pH the stability of CNT + DOX conglomerates remain stable.

## 3. Materials and Methods

### 3.1. Materials

Multi-walled carbon nanotubes (average outer diameter 9.5 nm, length about 1.5 μm) were purchased from Nanocyl S.A. They were produced via the Catalytic Chemical Vapor Deposition (CCVD) process and purified up to 95 wt.%. Doxorubicin hydrochloride (purity ≥ 98%) was obtained from Sigma Aldrich Russia, Moscow, Russian.

#### Modification of Multi-Walled Carbon Nanotubes

Multi-wall carbon nanotubes were modified via treatment in boiling nitric acid. The concentration of HNO_3_ was 8 mol/L, and the time of contact was 2 h. The ratio between the mass of MWCNTs and the HNO_3_ aqueous solution was 0.75 g on 0.10 L. The chemical modification of MWCNTs was repeated three times, and each time the mass decrease was 2.5 ± 0.2%. In the next step, the obtained sample was rinsed several times with water (until the supernatant reached pH between 6 and 7) and subsequently dried in air for 24 h at temperature 120 °C.

### 3.2. Methods

This work reports the experimental and molecular simulation studies of the doxorubicin adsorbed onto *m*MWCNTs. To characterize the properties of DOX onto *m*MWCNTs, such experimental methods as Fourier-Transform Infrared (FT-IR), UV-Vis Spectroscopy, Raman Spectroscopy, Transmission Electron Microscopy (TEM) and Scanning Electron Microscopy (SEM) were used. In addition, molecular simulations were also performed to complete the description of the release process of DOX from MWCNTs.

#### 3.2.1. FTIR

The samples in powder forms of *p*MWCNTs, *m*MWCNTs, pure DOX and DOX onto *m*MWCNTs were pelleted with KBr (a special chemical additive integrating a pill without affecting the physicochemical properties of a substance and transparent in the mid-infrared range). FTIR measurements were carried out in the mid-infrared range (4000–500 cm^−1^), with a resolution of 2 cm^−1^, on the EXACLIBUR 3000 spectrometer. 128 scans were done for the spectrum at room temperature.

#### 3.2.2. Raman Spectroscopy

Raman spectra were recorded by means of the InVia Ranishaw Raman Microscopy system (Ranishaw, Old Town, Wotton-under-Edge, UK) with the 633 nm He/Ne laser (0.75 mW laser power, Stage I) and 1800 g/mm grating. The laser light was focused on the sample with a 50×/0.75 microscope objective (LEICA). The measurements were also repeated using surface-enhanced Raman spectroscopy (SERS) with 785 nm diode laser (0.1 mW laser power, Stage I) while strengthening the signal with the gold substrate. All Raman spectra were obtained from 450 to 4000 cm^−1^ within 20 s acquisition time. The spectra were corrected by the WiRETM 3.3 software attached to the instrument. Measurements of peak positions were performed by means of the Lorentz profile at OriginPro 8.3 software (Northampton, MA, USA).

#### 3.2.3. Microscopic Methods

The analyzed systems were also characterized by microscopic methods. Scanning Electron Microscope (SEM) Zeiss Merlin was used to obtain the surface morphology. This microscope was equipped with Energy Dispersed Spectroscopy (EDS) system, which allowed one to carry out the chemical analysis on the studied surface. The details on the internal structure of the analyzed samples were obtained by Transmission Electron Microscope (TEM) Zeiss Libra 120 and High-Resolution Transmission Electron Microscope (HRTEM) Jeol ARM 200F.

#### 3.2.4. UV-VIS

UV-Vis spectroscopy measurements were carried out by UV-Vis spectrophotometer Shimadzu 2401 at λmax = 484 nm. The spectrophotometric standard curve method was used to determine the concentration of both adsorptions and released doxorubicin.

##### The Release Process of DOX from mMWCNTs

Quantification of released doxorubicin from carbon nanotubes was performed in one liquid medium. Due to the acidic pH of the tumour, the acetic buffer was chosen as the liquid medium at temperature 298 K to study the doxorubicin release process. In acidic pH, the major contribution to the total interaction energy of the drug–carrier system makes the Van der Waals interactions. The acidic buffer was prepared by dissolving 0.1 moles (8.2 g) of sodium acetate in 1 L of 0.1 M acetic acid solution. Ten repetitions of pH measuring were done, and the obtained results were in the range of pH = 4.95–5.03. The analyzed composite of DOX-MWCNTs was firstly obtained by using 100 mg of *m*MWCNTs which were suspended in the 10 mL of the solution of doxorubicin in phosphate-buffered saline (PBS) with concentration c = 40 mg/mL. This mixture was shaken for 24 h. The concentration of DOX after filtration of DOX-MWCNTs dropped to 22.5 μg/mL. This result indicated that 99.97% of DOX had been adsorbed onto MWCNTs. To quantify the release process of DOX from MWCNTs, four studies of this process were performed. In each of them the ca. 10 mg of DOX- MWCNTs material was placed in the acetic buffer. The volume of this medium varied between 10 mL and 100 mL, and the mixtures were shaken. The concentration of released DOX was determined by means of the spectrophotometric standard curve method.

##### Determination of Release Kinetics of Doxorubicin in Liquid Medium

Due to the acidic pH, the major feature of tumour tissue, the kinetics of DOX release was studied in the acetic buffer. The research was performed for seven mixtures of DOX-MWCNTs. In each one, ca. 10 mg of DOX-MWCNTs material was placed in the acetic buffer. The volume of this medium was 50 mL due to the highest release of DOX observed for this value of acetic buffer. The mixtures were shaken for 1, 3, 5, 10, 20 and 30 min. The kinetic models were used to evaluate the release mechanism of DOX from *m*MWCNTs. Therefore, the release kinetics of DOX was fitted to the selected models such as zero-order, first-order and second-order kinetics, Higuchi, Hixson–Crowell, Weibull, and Korsmeyer–Peppas/Ritger-Peppas (Power-law model).

##### Molecular Dynamics Simulations

Computer simulations were performed by the molecular dynamics (MD) method for the multi-walled and single-walled carbon nanotube systems with doxorubicin in water, that is, at neutral pH. Groeningen Machine for Chemical Simulations (GROMACS) [52] with OPLS-AA forcefield [53,54] was used for this purpose. The canonical ensemble (NVT) was applied at a constant temperature of 293.15 K at a v-rescale thermostat with a time constant for coupling (=0.1 ps). The PME-based algorithm was used to calculate electrostatic interactions. Constraining all bonds to the forcefield equilibrium lengths was carried out with the LINCS algorithm [55]. The time step was set to the value *dt* = 1 fs, ensuring good performance and keeping the system stable (prevents runway due to the too long timestep). The cutoff distance for the nonbonded interactions (Van der Waals and electrostatic) was set to the default value for OPLS-AA forcefield, that is, to 1.0 nm. The potential-shift modifier was chosen for these interactions. Limitations due to the computational complexity of the problem in relation to the performance of modern computing machines, even supported by graphics cards for molecular computing, forced us to simulate small systems against experimental real-scale systems. Moreover, real systems showed the inhomogeneity of certain parameters, such as the density of a given type of particles within the system. In the case of numerical models, due to, among other things, the finite and small size of the systems, we had to set these parameters precisely. We fixed the initial average density of both the nanotubes and the DOX. The size of the cuboid simulation box was chosen to 20 × 80 × 20 nm. The nanotubes were modeled as three-wall zigzag carbon nanotubes (MWCNTs) and single-wall zigzag carbon nanotubes (SWCNTs). Each SWCNTs, including those which built MWCNTs, was generated by means of the buildCstruct script [56] adapted to our needs. Nanotube diameters were chosen to match those of the experiment: *f*1 = 6.42 nm, *f*2 = 7.12 nm, *f*3 = 7.83 nm for MWCNT and *f* = 6.42 nm for SWCNT. The length of nanotubes was set to *L* = 20 nm. It was chosen to be long enough to observe the phenomena we were interested in and short enough to get reasonable simulation time, which, in total, we count in months anyway. The number of DOX molecules was chosen to keep the mass ratio between DOX and MWCNTs as in the optimal adsorption case in the experiment for modified MWCNTs, that is (mDOX/mCNTs) = 4/1.

## 4. Conclusions

We have reported the experimental and simulation studies of the release process of doxorubicin from modified carbon nanotubes. The surface of MWCNTs has been modified to obtain the more efficient drug adsorption/desorption on/from this carrier. Our studies have shown the relatively rapid release process of DOX from *m*MWCNTs at the acidic pH, where almost 90% of the drug has been released after 20 min. To describe the mechanism of release, we used several kinetic models. We have found that the experimental data well fitted to the Korsmeyer–Peppas/Ritger-Peppas model. Based on kinetics parameters, we can conclude that the release process of DOX from the hydrophobic surface of MWCNTs was controlled by Fickian diffusion.

Complementary studies by molecular dynamics simulations at neutral pH, i.e., at a pH close to that of blood, have shown in turn the stability of adsorbed doxorubicin aggregates on nanotubes as well as the stability of free doxorubicin aggregates, leading to the conclusion that no resorption process occurs at neutral pH.

## Figures and Tables

**Figure 1 ijms-22-12003-f001:**
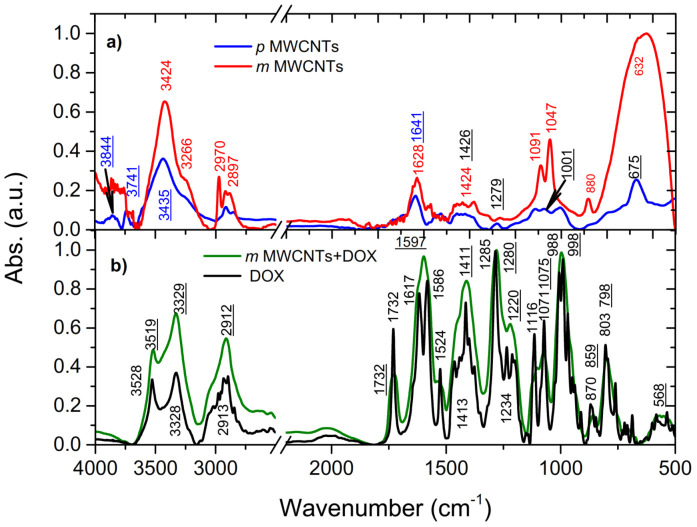
FTIR comparison of: *p*MWCNTs and *m*MWCNTs (**a**) and DOX and DOX onto *m*MWCNTs (**b**).

**Figure 2 ijms-22-12003-f002:**
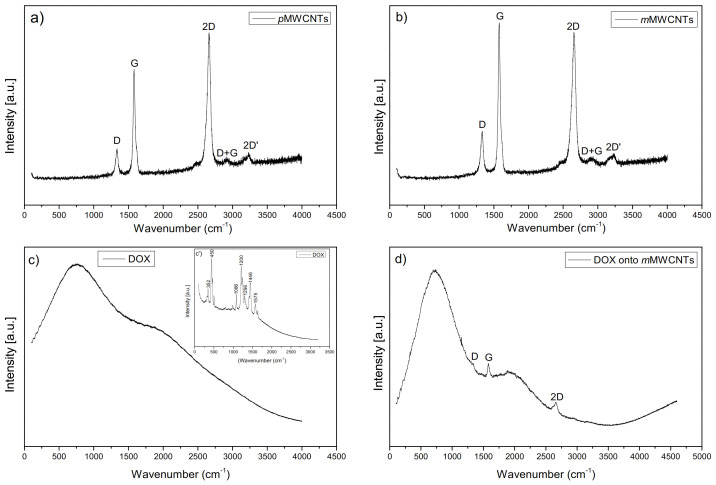
Raman spectroscopy results for *p*MWCNTs (**a**), *m*MWCNTs (**b**), DOX (**c**,**c’**) and DOX onto *m*MWCNTs (**d**).

**Figure 3 ijms-22-12003-f003:**
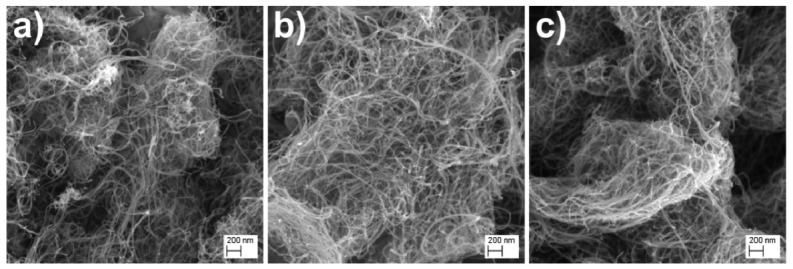
SEM images of (**a**) *p*MWCNTs (**b**) *m*MWCNTs and (**c**) *m*MWCNTs with adsorbed DOX.

**Figure 4 ijms-22-12003-f004:**
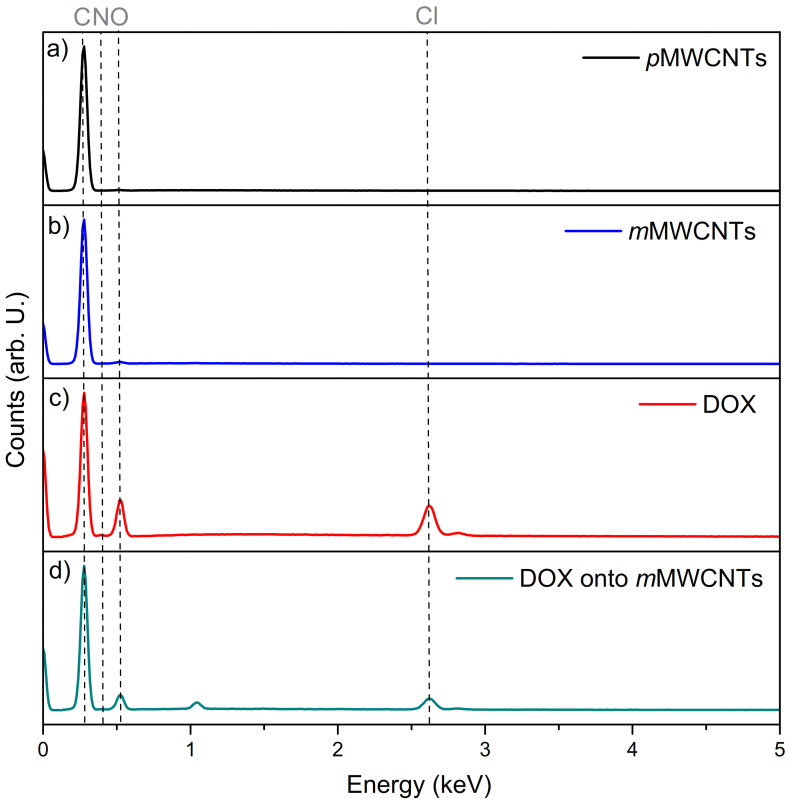
EDS spectra of *p*MWCNTs (**a**), *m*MWCNTs (**b**) doxorubicin (**c**) and doxorubicin adsorbed onto *m*MWCNTs (**d**).

**Figure 5 ijms-22-12003-f005:**
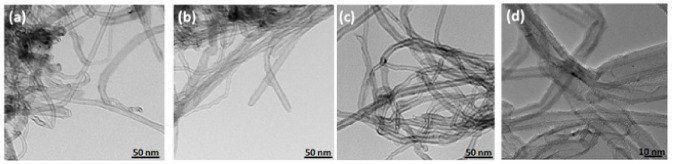
TEM images of pristine MWCNTs (**a**), chemically modified MWCNTs (**b**) and chemically modified MWCNTs with adsorbed doxorubicin (**c**,**d**).

**Figure 6 ijms-22-12003-f006:**
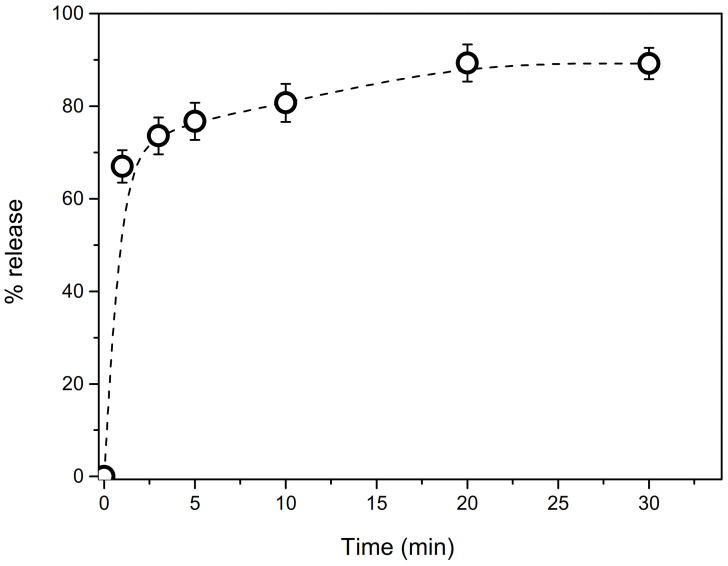
Release kinetics curve of doxorubicin from DOX-MWCNTs. Data are presented as mean ± SD (*n* = 3).

**Figure 7 ijms-22-12003-f007:**
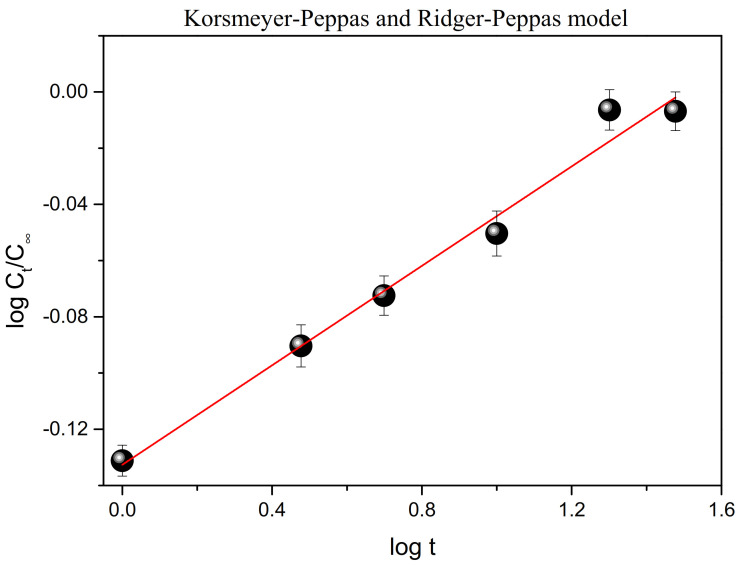
Power-law model for release process of DOX from *m*MWCNTs. Data are presented as mean ± SD (*n* = 3).

**Figure 8 ijms-22-12003-f008:**
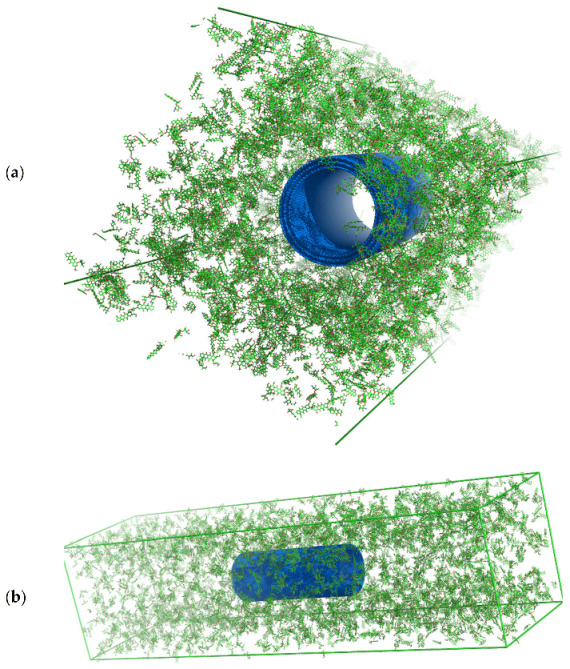
Snapshots of the input configuration for the in vacuo simulation step for MWCNTs (indicated in blue) with doxorubicin (DOX, molecules indicated in green/dark red elements) from two perspectives: (**a**) view to see the nanotube through; (**b**) overall view of the entire simulation box.

**Figure 9 ijms-22-12003-f009:**
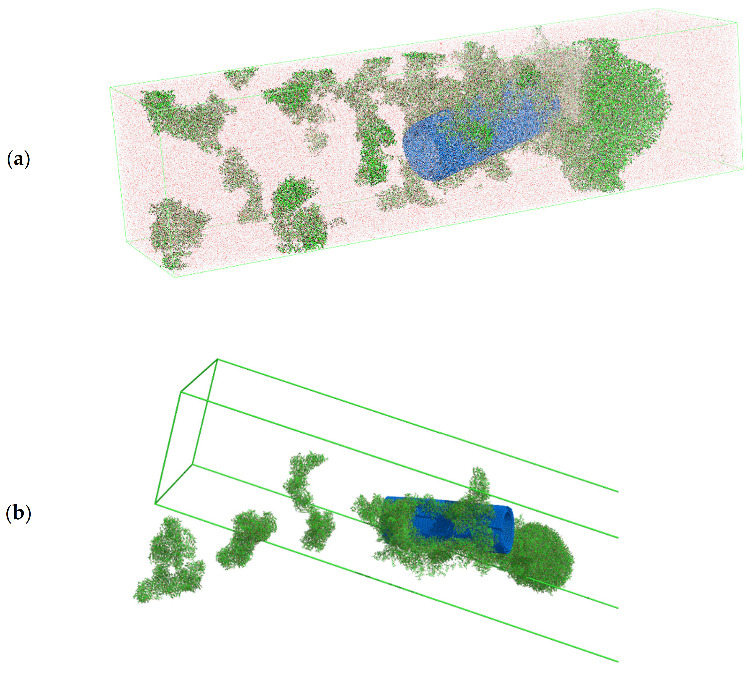
Molecular dynamics simulation snapshots of MWCNTs (marked in blue) system with doxorubicin (DOX, molecules indicated in green/dark red elements) and water (light red, high transparency for clarity) after time t = 35 ns from two perspectives: (**a**) view of the whole system in coordinates collapsed into the simulation box (periodic boundary); (**b**) view of the system in expanded coordinates, water molecules are not shown for transparency, four aggregates visible, three smaller DOX-only and one MWNTs + DOX.

**Figure 10 ijms-22-12003-f010:**
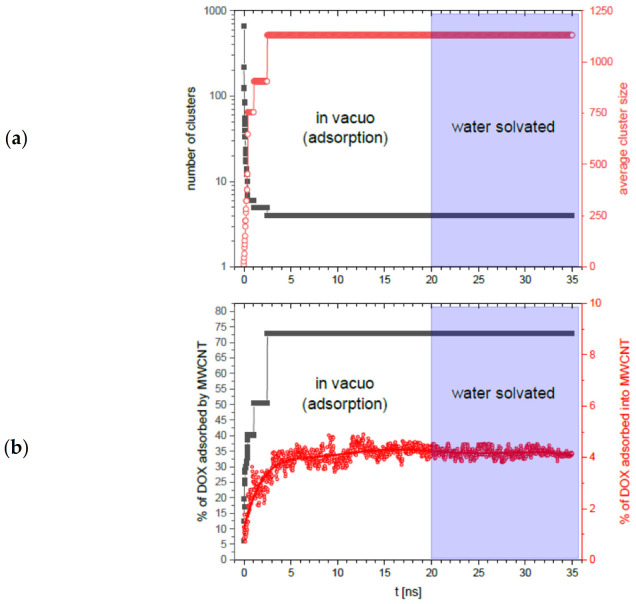
Results of molecular dynamics simulation of MWCNTs + DOX system. Selected parameters as a function of time, t, during the adsorption process in vacuo (from *t* = 0 to *t* = 20 ns) followed by the simulation of the system in solution with water (from *t* = 20 ns to *t* = 35 ns): (**a**) number of aggregates (grey squares) and average aggregate size (red circles); (**b**) total percentage of DOX molecules adsorbed by MWCNTs (grey squares) and percentage of DOX molecules adsorbed inside MWCNTs (red circles).

**Figure 11 ijms-22-12003-f011:**
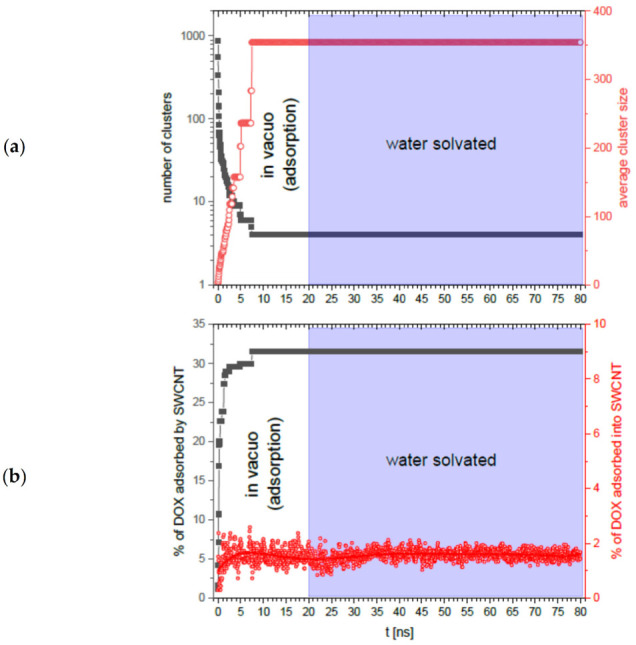
Selected physical quantities as a function of time, *t*, from molecular dynamics simulation of SWCNTs + DOX system during adsorption in vacuo (from *t* = 0 to *t* = 20 ns), and subsequent simulation of the system in solution with water (from *t* = 20 ns to *t* = 80 ns): (**a**) number of aggregates (grey squares) and average aggregate size (red circles); (**b**) total percentage of DOX molecules adsorbed by SWCNTs (grey squares) and percentage of DOX molecules adsorbed inside SWCNTs (red circles).

**Table 1 ijms-22-12003-t001:** Results on doxorubicin release in the equilibrium mode.

Acetic Buffer Volume (mL)	% Release
10	73.8
20	79.8
50	88.2
100	86.9

**Table 2 ijms-22-12003-t002:** List of parameters obtained from the kinetic models release of doxorubicin from *m*MWCNTs.

Kinetic Model	Parameters	Values
The zero-order	*K*_0_/g mg^−1^ min^−1^	0.0829
	R^2^	0.223
The first-order	*K*_1_/min^−1^	0.0552
	R^2^	0.660
The second-order	*K*_2_/mL μg^−1^ min^−1^	0.0015
	R^2^	0.892
Higuchi	*K_H_*/mg mL^−1^ min^−1/2^	34.9
	R^2^	0.840
Hixson-Crowell	*K_H_*_-*C*_/μg^1/3^ mL^−1/3^ min^−1^	0.0630
	R^2^	0.546
Weibull	B	0.3612
	R^2^	0.877
Power law	*K*	0.7379
	*n*	0.088
	R^2^	0.979

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
