# Peer review of "Description of Release Process of Doxorubicin from Modified Carbon Nanotubes"

_ijms, 2021, doi:10.3390/ijms222112003_

Round 1

Reviewer 1 Report

The manuscript entitled "Description of the release process of doxorubicin from modified carbon nanotubes" provides sound discussion with a reasonable set of experiments. Indeed, the manuscript needs to be revised considering the following suggestions.

  1. Abstract need to be modified incorporating novelty and key finding of the current investigations. The current research gap in doxorubicin release through multi-walled carbon nanotube should be highlighted and mention how the current investigation is helpful to fulfil this research gap.
  2.  The introduction section needs to be modified and remove the generalized information. This section should be mainly focused on the drug release perspectives through carbon nanotubes (single/multi-walled carbon nanotube). Incorporate the doxorubicin release through carbon nanotube already available in the literature and highlight the current research gap. Mention current study design how helpful to fulfill this research gap particularly drug release perspectives through a carbon nanotube. Highlight the significance of molecular dynamic simulation investigation. 
  3. title of section 3.2. changed to "characterization of multi-walled carbon nanotubes
  4. The experimental setup description for the drug release process is in section 3.2.4.1. need to be elaborated further. Particularly description of media composition, media pH, media temp, description of drug release apparatus.... etc.
  5. The author should also mention the drug release behavior at blood pH. What will be the release behavior after in vivo administration of this doxorubicin carbon nanotube system? The drug release media should be bio-relevant and mimic/simulate to in vivo system.
  6. The author has to mention the various factors that influence the release profile of doxorubicin in the current investigation. It should be shown through a suitable experimental design.  
  7. The hypothesis of the present experimental design is mainly based on doxorubicin release behavior through multi-walled carbon nanotube but in my opinion, detailed/comprehensive investigation missing particularly which help to add on the information related to doxorubicin release behavior through multi-walled carbon nanotube already available in the literature.  

Author Response

  1. Abstract need to be modified incorporating novelty and key finding of the current investigations. The current research gap in doxorubicin release through multi-walled carbon nanotube should be highlighted and mention how the current investigation is helpful to fulfil this research gap.

 Response:

The abstract has been corrected. All changes made were marked in yellow.

  1. The introduction section needs to be modified and remove the generalized information. This section should be mainly focused on the drug release perspectives through carbon nanotubes (single/multi-walled carbon nanotube). Incorporate the doxorubicin release through carbon nanotube already available in the literature and highlight the current research gap. Mention current study design how helpful to fulfill this research gap particularly drug release perspectives through a carbon nanotube. Highlight the significance of molecular dynamic simulation investigation.

 Response:

As the Reviewer suggested, the Introduction has been corrected. All changes were marked in yellow.

  1. Title of section 3.2. changed to "characterization of multi-walled carbon nanotubes

 Response:

The title of section 3.2. was changed to “Methods used for characterization of multi-walled carbon nanotubes”

  1. The experimental setup description for the drug release process is in section 3.2.4.1. need to be elaborated further. Particularly description of media composition, media pH, media temp, description of drug release apparatus.... etc.

 Response:

Information about aperture was written in section 3.2.4 (highlighted in yellow). The additional description of the conditions of the experiment was added in the text in section 3.2.4.1 (highlighted in yellow):

  1. The author should also mention the drug release behavior at blood pH. What will be the release behavior after in vivo administration of this doxorubicin carbon nanotube system? The drug release media should be bio-relevant and mimic/simulate to in vivo system.

 Response:

Additional information were added in Abstract, as well as in section 2.5. Molecular dynamics simulations and Conclusions. All changes made were marked in yellow.  In vivo studies is a further stage of our research.

  1. The author has to mention the various factors that influence the release profile of doxorubicin in the current investigation. It should be shown through a suitable experimental design.

Response:

We added an apiece of information in section 2.4. UV-VIS spectroscopy (highlighted in yellow)

  1. The hypothesis of the present experimental design is mainly based on doxorubicin release behavior through multi-walled carbon nanotube but in my opinion, detailed/comprehensive investigation missing particularly which help to add on the information related to doxorubicin release behavior through multi-walled carbon nanotube already available in the literature.

 Response:

We have added information related to release behavior in part 2.4 UV-VIS spectroscopy (highlighted in yellow)

Reviewer 2 Report

Dear Authors,

In my opinion, the novelty of the manuscript is high and so it may be published after minor revision.

The list of the most important comments is as follows:

- In the abstract, more details should be given, e.g.: buffers pH in the release study, the modification of nanotubes,

- I don't understand the use of acidic pH, the pH of tissues is 7.4, please explain,

- I don't need the information from lines 30-45,

- The release of DOX should be more than 30 minutes, please explain,

- In the next stage of the study, the action of nanoparticles with DOX on both physiological and pathological cells should be determined,

- It should be noted that DOX HCl has been used from in the abstract and in the introduction.

Best regards,

Reviewer

Author Response

- In the abstract, more details should be given, e.g.: buffers pH in the release study, the modification of nanotubes,

Response:

The abstract has been corrected, all changes made were marked in yellow.

- I don't understand the use of acidic pH, the pH of tissues is 7.4, please explain,

Response:

In the Experimental part, we used acidic pH due to the acidic pH of the tumor. In acidic pH, the carboxylic groups on MWCNTs surface are protonated. The neutral pH – 7.4, which is the pH of tissues was used in Molecular Dynamic simulations. The results of MD calculations have shown that in pH 7.4 the system is stable and the release process does not occur. This result can be explained by stronger hydrogen bonding interaction at neutral pH.

- I don't need the information from lines 30-45,

Response:

The Introduction has been corrected. All changes were marked in yellow.:

 - The release of DOX should be more than 30 minutes, please explain,

Response:

The rapid release of DOX is due to the way doxorubicin binds to the nanotube and other DOX molecules through hydrogen bonding. The acidic pH causes the weakening of the hydrogen bonds and, consequently, the rapid release kinetics. DOX, covalently bonded to the nanotube, is released for a much longer time. The experiments were carried out for 2h, however, due to lack of changes only first  30 min were presented in manuscript.

- In the next stage of the study, the action of nanoparticles with DOX on both physiological and pathological cells should be determined,

Response:

This is a further stage of our research.

- It should be noted that DOX HCl has been used from in the abstract and in the introduction.

 Response:

The abstract and introduction have been corrected, changes were marked in yellow.

Reviewer 3 Report

In this manuscript the authors aim at presenting an investigation into the rate kinetics of the release of Dox from MWCNTS

I have several comments

  1. No information about the chemical modification of the CNTs with Dox is mentioned
  2. What is the mechanism of chemical attachment-is it using the OH or the NH2 group?
  3. How was the loading content calculated? UV-Vis or HPLC?
  4. The investigation was performed only at pH 7.4 , other pH levels should be included
  5. NMR data is lacking
  6. FRIT spectra of Dox + MWCNTS does not show any bond formation, usually an amide/ester linkage should appear
  7. SEM images of SWCNT and Dox+ MWCNT do not show any visible changes. Is it possible to get image of Dox+ MWCNT after loading and after release?
  8. In Fig 4, the authors say the Dox is adsorbed on to the MWCNTs , this is a physical phenomenon and not a chemical modification as is claimed throughout the manuscript.
  9. From Figure 6, it appears as if all the drug is released within an hour at the physiological pH. This is undesirable as this causes off target toxicity. The tumor pH is lower, and a differential release profile is preferred. Is this a cumulative% release? The same study should be carried out at different pH
  10. Replicates/standard deviation is missing
  11. The following references must be added:

https://link.springer.com/article/10.1007/s11051-018-4239-x

https://pubs.acs.org/doi/abs/10.1021/acs.molpharmaceut.0c00963

https://pubmed.ncbi.nlm.nih.gov/30503561/

Author Response

  1. No information about the chemical modification of the CNTs with Dox is mentioned

Response:

MWCNTs were modified via treatment in boiling HNO3. More information about this process is presented in section 3.1.1

  1. What is the mechanism of chemical attachment-is it using the OH or the NH2 group?

Response:

Both groups are able to create covalent bond.

  1. How was the loading content calculated? UV-Vis or HPLC?

Response:

The adsorption process was determined by using the UV-Vis method.

  1. The investigation was performed only at pH 7.4 , other pH levels should be included

Response:

Theoretical calculations carried out at pH=7 show that the release does not occur. It can be explained by stronger hydrogen bonding interaction at neutral and based pH. For that reason, experiments were performed for pH = 5.4 which is typical for the tumour microenvironment.

  1. NMR data is lacking

Response:

The chemical composition (properties) of the system was determined using such methods as EDS, Raman spectroscopy and FTIR. We agree that the NMR method would be very useful.

  1. FRIT spectra of Dox + MWCNTS does not show any bond formation, usually an amide/ester linkage should appear

Response:

DOX molecules containing polar functional groups, i.e., hydroxyl group COH, carbonyl group C = O, amino group NH2, which are mainly involved in intermolecular interactions, can form more stable bonds by attaching to functional groups (COOH and OH) on MWCNTS. However, functional groups of DOX molecules take part in intermolecular interactions (and form a large number of hydrogen bonds of different strengths). Observed phenomena resemble DOX aggregation on MWCNTs. As a result, created covalent bonds are a minority, and a large number of different hydrogen bonds cover the characteristic of an amide and ester band.

We decide to add some additional text in section 2.1. Fourier Transform Infrared (FTIR) Spectroscopy to clarify the inaccuracy. All added text has been marked in yellow.

  1. SEM images of SWCNT and Dox+ MWCNT do not show any visible changes. Is it possible to get image of Dox+ MWCNT after loading and after release?

Response:

We agree that differences between MWCNTs and MWCNTs with adsorbed DOX are slightly visible in the SEM pictures. We did not carry out the SEM experiments after DOX release because this process was described using the UV-Vis method. Unfortunately, currently, we do not have the possibility to do SEM experiments for the studied system after DOX release.

  1. In Fig 4, the authors say the Dox is adsorbed on to the MWCNTs , this is a physical phenomenon and not a chemical modification as is claimed throughout the manuscript.

Response:

The aim of the chemical modification of MWCNTs was to improve their dispersion and stability as well as allow to attach of Dox through the formation of covalent bonds. However, the main force of DOX adsorption on MWCNTs was hydrogen bonding. It is worth noticing that adsorption on pristine (not modified) MWCNTs is marginal. It confirms the strong influence of chemical modification on adsorption and release tendency.

  1. From Figure 6, it appears as if all the drug is released within an hour at the physiological pH. This is undesirable as this causes off target toxicity. The tumor pH is lower, and a differential release profile is preferred. Is this a cumulative% release? The same study should be carried out at different pH

Response:

The experimental studies were conducted at acidic pH. Complementarily, the research was also carried out at neutral pH with the use of MD simulations. As can be seen from the obtained results, about 80% of DOX is released at acidic pH, which seems to be a desirable result, while at neutral pH, the system remains stable and the release process of DOX is not observed.

  1. Replicates/standard deviation is missing

Standard deviation values was added.

  1. The following references must be added:

https://link.springer.com/article/10.1007/s11051-018-4239-x

https://pubs.acs.org/doi/abs/10.1021/acs.molpharmaceut.0c00963

https://pubmed.ncbi.nlm.nih.gov/30503561/

Response:

The references were added in the introduction section.

Round 2

Reviewer 1 Report

The revised manuscript (ijms-1433926) improved well. In my opinion, the present manuscript should be considered for publication in IJMS (ISSN 1422-0067). 

Reviewer 3 Report

The authors have tried to respond to most questions and clarified their results. This can be accepted for publication